# In Vitro Cytotoxic Effects and Mechanisms of Action of Eleutherine Isolated from *Eleutherine plicata* Bulb in Rat Glioma C6 Cells

**DOI:** 10.3390/molecules27248850

**Published:** 2022-12-13

**Authors:** Victoria Mae Tsuruzaki Shinkai, Izana Marize Oliveira Sampaio, Eline Gomes dos Santos, Adan Jesús Galué-Parra, Dionisia Pelaes Ferreira, Drielly Dayanne Monteiro Santos Baliza, Neidiane Farias Ramos, Raphael Sanzio Pimenta, Rommel Mario Rodriguez Burbano, Chubert Bernardo Castro Sena, Barbarella Matos Macchi, Irlon Maciel Ferreira, Edilene Oliveira Silva, José Luiz Martins do Nascimento

**Affiliations:** 1Programa de Pós-Graduação em Farmacologia e Bioquímica, Instituto de Ciências Biológicas, Universidade Federal do Pará, Belem 66075-110, Brazil; 2Laboratório de Neuroquímica Molecular e Celular, Instituto de Ciências Biológicas, Universidade Federal do Pará, Belem 66075-110, Brazil; 3Laboratorio de Biologia Estrutural, Instituto de Ciências Biológicas, Universidade Federal do Pará, Belem 66075-110, Brazil; 4Programa de Pós-Graduação em Ciências Farmacêuticas, Departamento de Ciências Biológicas e da Saúde, Universidade Federal do Amapá, Macapa 68902-280, Brazil; 5Laboratório de Biocatálise e Síntese Orgânica Aplicada, Departamento de Ciências Exatas e Tecnológicas, Universidade Federal do Amapá, Macapa 68902-280, Brazil; 6Programa de Pós-Graduação em Biodiversidade e Biotecnologia da Região Norte, Universidade Federal do Tocantins, Palmas 77001-090, Brazil; 7Laboratório de Citogenética Humana, Instituto de Ciências Biológicas, Universidade Federal do Pará, Belem 66075-110, Brazil; 8Instituto Nacional de Ciência e Tecnologia em Neuroimunomodulação (INCT-NIM), Rio de Janeiro 21040-900, Brazil; 9Instituto Nacional de Ciência e Tecnologia de Biologia Estrutural e Bioimagem (INCT-INBEB), Rio de Janeiro 21941-902, Brazil

**Keywords:** glioma C6, rat brain tumor, *Eleutherine plicata*, eleutherine, cytotoxicity

## Abstract

Gliomas are the most common primary malignant brain tumors in adults, and have a poor prognosis, despite the different types of treatment available. There is growing demand for new therapies to treat this life-threatening tumor. Quinone derivatives from plants have received increased interest as potential anti-glioma drugs, due to their diverse pharmacologic activities, such as inhibiting cell growth, inflammation, tumor invasion, and promoting tumor regression. Previous studies have demonstrated the anti-glioma activity of *Eleutherine plicata*, which is related to three main naphthoquinone compounds—eleutherine, isoeleutherine, and eleutherol—but their mechanism of action remains elusive. Thus, the aim of this study was to investigate the mechanism of action of eleutherine on rat C6 glioma. In vitro cytotoxicity was evaluated by MTT assay; morphological changes were evaluated by phase-contrast microscopy. Apoptosis was determined by annexin V–FITC–propidium iodide staining, and antiproliferative effects were assessed by wound migration and colony formation assays. Protein kinase B (AKT/pAKT) expression was measured by western blot, and telomerase reverse transcriptase mRNA was measured by quantitative real-time polymerase chain reaction (qRT-PCR). Eleutherine reduced C6 cell proliferation in a dose-dependent manner, suppressed migration and invasion, induced apoptosis, and reduced AKT phosphorylation and telomerase expression. In summary, our results suggest that eleutherine has potential clinical use in treating glioma.

## 1. Introduction

Glioma is a fatal malignancy derived from the glial cell lineage with a high recurrence rate. Although less prevalent than other neoplasms, 60% are aggressive high-grade gliomas (grade IV), which migrate and strongly infiltrate the brain parenchyma [1,2,3,4,5]. Gliomas, in general, are responsible for approximately 38% of primary brain tumors, and represent more than 70% of all central nervous system (CNS) tumors, the most frequent and malignant being glioblastoma multiforme [6,7]. Glioblastomas have the highest proportion of malignancy and the worst prognosis of CNS tumors. Only 5.5% of patients usually survive for 5 years after diagnosis [8] and have a 1-year survival rate because of the limitations of treatment’s approaches [9].

Treatment strategies for glioma differ, depending on tumor size, location, and distinct tumor subtype, and involve a combination of surgery, radiotherapy, and chemotherapy. Typically, glioma chemotherapy includes the antibody nimotuzumab and temozolomide, but their efficacy is low, and these tumors have poor prognosis [10,11,12,13,14]. In addition, gliomas develop resistance to numerous drugs [15,16,17,18,19]. Therefore, there is a real need for new antitumor agents to treat this deadly cancer, and efforts are ongoing to identify more efficient and active molecules to regulate the tumor’s progression.

Several plants, such as *Eleutherine plicata*, *Eleutherine americana*, and *Cipura paludosa*, are important sources of chemicals and contain high levels of naphthoquinones, a class of natural products with diverse biological activities. Aqueous extracts of the dried bulbs of these herbs have been described to generate oxidative stress by inducing the deleterious endogenous formation of a bioactive oxygen-derived species that inhibits inflammation, promotes apoptosis, and exhibits important anticancer properties in different cancer cell lines, such as glioma (U-251), breast (MCF-7), ovary (NCI/ADR-RES), kidney (786-0), lung non-small cell (NCI-H460), colon (HT-29), HepG2 cells, and leukemia (K562) [20,21].

These effects are mainly related to the presence of naphthoquinones, more specifically eleutherine, isoeleutherine, and eleutherol (Figure 1) [20,22]. Previous works have demonstrated the antiproliferative effect of eleutherine and isoeleutherine isolated from a methanolic extract of *Cipura paludosa* bulbs in glioma (U251) and breast cancer (MCF-7) lines [20]. The results indicate that eleutherine is more cytotoxic than isoeleutherine. These compounds are epimeric isomers and have a 1,4-naphthoquinone moiety with only one structural difference, the β-methyl group of eleutherine and the α-methyl group of isoeleutherine. Thus, it follows that the higher activity of eleutherine is related to the chirality of its pyran ring with the β-methyl group [20].

Some studies suggest that naphthoquinones may exert their regulatory activities in cancer cells by acting on different signal transduction pathway family proteins, such as phosphatidylinositol-3-kinase/AKT and MAP kinase (MAPK) [23,24,25]. In addition, it is known that eleutherine reversibly inhibits the catalytic activity of human topoisomerase II by stabilizing the DNA–enzyme complex in the presence of ATP [26]. Thus, by inhibiting the activity of this enzyme, eleutherine can cause transient double-strand breakage in DNA and contribute to biological oxidative processes [27,28].

These activities open new opportunities for pharmacotherapy, especially for cancer [20,29,30]. However, eleutherine’s mechanism of action remains elusive. Thus, isolated eleutherine was used in this work to elucidate a possible mechanism of its action on C6 glioma cells.

## 2. Results

### 2.1. Characterization of the Compound Isolate

The compound isolated from *Eleutherine plicata* bulbs was obtained as a brownish yellow crystal. Starting from 3 g of the lyophilized ethanolic extract yielded over 49 mg of the pure compound, with 95% relative concentration (Figure 2, see Appendix A), with MS (70 eV, EI) 272 (42), 257 (100), 243 (52), 214 (32),157 (10), and 121 (25), and the probable molecular formula was deduced as C_14_H_12_O_4_. The melting point was of 144–146 °C. The IR spectrum showed the presence of characteristic peaks, such as -C-H_sp2_ (2974 cm^−1^), -C-H_sp3_ (2913), -C=O (1777 cm^−1^), of benzene ring (1582cm^−1^) absorptions. The molecular characterization data were compared with those in the literature [31].

### 2.2. Cytotoxic Effect of Eleutherine Treatment on C6 Cells

Eleutherine presented a cytotoxic effect on C6 cells. After 6 and 12 h of treatment, cell death was observed at 1 μM with IC_50_ of 32.33 μM (±1.25) and 28.46 μM (±1.85), respectively (Figure 3A,B). During 24 h of treatment, a cytotoxic effect was observed at a concentration range of 0.05–100 μM, with an IC_50_ of 4.98 μM (±0.22) (Figure 3C). Thus, eleutherine reduced cell viability in a concentration- and time-dependent manner (Figure 3D).

### 2.3. Morphological Changes in C6 Cells after Eleutherine Treatment

No structural changes were observed in the control, which was characterized by the presence of fusiform cells and monolayer cells. Cells treated with eleutherine at 1 μM presented small morphological changes. However, cells treated at 20 and 40 µM showed structural changes characteristic of apoptosis, such as large and round cells and irregularly shaped cells with cytoplasmic shrinkage (white arrow). Treatment with eleutherine at 100 μM resulted in cell death (black arrow) (Figure 4).

### 2.4. Eleutherine Reduces Colony Formation by C6 Cells

Treatment with eleutherine at 20, 40, and 100 μM for 12 h resulted in decreases in colony formation (52.44%, 94.52%, and 99.18%, respectively) compared with that observed in the untreated group. No reduction in colony formation was observed at 1 µM eleutherine (Figure 5).

### 2.5. Eleutherine Induces C6 Cell Apoptosis

The annexin V–PI assay was used to confirm apoptotic cell death caused by eleutherine. The C6 cells were treated with different concentrations of eleutherine (1 μM, 20 μM, and 40 μM) and displayed increased numbers of apoptotic cells, mainly in the late stages of apoptosis, as follows: at 1 μM, 57%; at 20 μM, 44.4%; and, at 40 μM, 77%, compared with the control and positive control (camptothecin), as shown in Figure 6. Our results demonstrate that eleutherine induces apoptosis in a dose-dependent manner to a level higher than that observed in the positive control.

### 2.6. Eleutherine Reduces the Expression of pAKT in C6 Cells

The expression of phosphorylated AKT was reduced after treatment with eleutherine by 48.84% at 20 μM and by 39.27% at 40 μM (Figure 7). Thus, the PI3K/AKT pathway in the glioma cells must be inhibited by the action of eleutherine.

### 2.7. Eleutherine Reduces Telomerase (TERT) mRNA Expression

Eleutherine treatment reduced TERT expression in a dose-dependent manner at 1 μM (17.1%), 20 μM (37.3%), and 40 μM (45.8%). The results suggest that after treatment of glioma cells with eleutherine there was a loss of telomerase activity that may have induced cell death (Figure 8).

## 3. Discussion

Naphthoquinones are significantly cytotoxic to different types of tumor cell lines. Recent evidence indicates that the antiproliferative activity of *Eleutherine plicata* is due the presence of the three main naphthoquinones, namely eleutherine, isoeleutherine, and eleutherol [21]. Treatment of C6 cells with eleutherine resulted in a dose- and time-dependent change in cell viability. As expected, the IC_50_ decreased with the treatment time, reaching 4.98 μM in cells exposed up to 24 h, demonstrating the effectiveness and potency of eleutherine. These findings are similar to those obtained with other naphthoquinone derivatives in primary cultures of cancer cells [30]. Eleutherine and isoeleutherine isolated from the bulbs of *C. paludosa* exhibited promising cytotoxicity against glioma (U-251), with IC_50_ values between 2.6 and 13.8 mg/mL [20].

The IC_50_ value obtained in our experiment is comparable to those reported in the literature. In addition, it was possible to verify that eleutherine does not have cytotoxic effects on normal glial cells (see Appendix A), thus, proving its therapeutic potential and its safety in regard to healthy cells. From these results, we highlight the significant cytotoxic effect of eleutherine on C6 cells reported here.

Cancer cells, such as glioma cells, cause the failure of efficient cellular response and disable the inhibition of apoptosis. Our data indicate that eleutherine was able to induce apoptosis in C6 cells. Treatment resulted in increases in the apoptosis rates when analyzed by FITC-Annexin V/PI staining. The C6 cells treated with 40 μM eleutherine showed an increased rate of apoptosis, similar to a previous study with another naphthoquinone [32,33]. These data support the hypothesis that there is a correspondence between the IC_50_ and the inhibition of proliferation and a high apoptosis rate.

Phase-contrast microscopy revealed structural changes produced by eleutherine. Cells treated with 20 μM, 40 μM, and 100 μM eleutherine retracted, decreased in size, and underwent significant cytoplasmic shrinkage, similar to cells undergoing apoptosis. These results clearly indicate that decreased cell size and cytoplasmic shrinkage may be related to cell death [34].

Moreover, through the colony formation assay, it was possible to observe the antiproliferative effect of eleutherine on cultured cells. The cells treated with 20 μM eleutherine formed significantly fewer colonies, and treatment with 40 μM and 100 μM eleutherine almost completely inhibited the formation of colonies. Interestingly, the ability to form colonies is a sensitive indicator of undifferentiated cancer stem cells [35,36]. These results indicate that eleutherine can act on stem cells, a therapeutic strategy which prevents cancer from relapsing.

Treatment with eleutherine inhibited the migration of glioblastoma cells in comparison with the untreated control group and the vehicle group. With 1 μM eleutherine, it was possible to observe a reduction in the proliferation of cells and, at 20 μM, the cells showed changes in morphology, suggesting apoptosis. The morphology was totally altered in a dose-dependent way at 12, 24, and 48 h. These findings corroborate the results of the cell viability assay, highlighting the considerable antitumor potential of eleutherine, similar to other naphthoquinones, in an in vitro model of glioblastoma [23,24,25,32,33,37,38].

The AKT pathway is important in the genesis of several types of cancer, being overexpressed and playing critical roles in the survival, proliferation, invasion, and migration of cancer cells. This pathway is vital in the development of glioma cells and related to metabolism, epithelial–mesenchymal transition, and angiogenesis, allowing the glioma to acquire a high degree of invasiveness and malignancy by bypassing the mechanisms of apoptosis [37,39,40,41]. In our study, we found that eleutherine decreased the expression of p-Akt and inhibited the PI3K/AKT pathway, resulting in glioblastoma cell death.

It is known that *TERT* is a component of telomerase, a reverse transcriptase ribonucleoprotein complex that maintains telomere length in cells with high proliferative ability, and that it plays a key role in cancer formation. Telomere maintenance is affected by *TERT* gene amplification and epigenetic changes, such as DNA methylation, *TERT* promoter germ line and somatic mutations, and *TERT* structural variants [42,43]. Catalytic inhibitors of topoisomerases act either by inhibiting the binding of the enzyme to DNA or by preventing it from cleaving the DNA, compromising its cellular repair capacity [44,45]. Our results suggest that eleutherine, as with other naphthoquinones, reduces the telomerase activity that is critical to glioma cells growth [38].

More recent studies confirm the interaction of the telomerase with various intracellular signaling pathways including the PI3K/AKT/mTOR pathway, which mainly participates in inflammation, the epithelial-to-mesenchymal transition, and tumor cell invasion and metastasis [46,47]. Thus, a decrease in phosphorylated AKT expression inhibits hTERT expression, thereby reducing proliferative capacity, altering the cell cycle, and promoting apoptosis in glioma cells.

The results indicate that eleutherine treatment reduced the expression of phosphorylated AKT and *TERT* in a dose-dependent manner. These mechanisms directly influence the cell cycle and cells’ ability to undergo apoptosis, which are of great interest in research on new antineoplastic therapies.

## 4. Materials and Methods

### 4.1. Plant Material

The specimens of *Eleutherine plicata* used in this study were obtained from a private medicinal plant plot in the city of Palmas, Tocantins, Brazil. The exicata is deposited in the Herbarium of the Federal University of Tocantins under the code number 8214.

### 4.2. Production of Raw Extract

For the preparation of extracts, *Eleutherine plicata* of an average size of 5 cm were ground in a blender with 2 mL of distillated water and subjected to decoction for 2 min, under agitation, according to Baliza et al. (2022) [48]. The mixture was distributed in extraction funnels, and hexane was added to the desired final volume. The funnel mixture was homogenized and allowed to sit until phase separation occurred. The hexane fraction was subjected to rotary evaporation to concentrate the extract in the flask. The flask was washed with hexane to loosen the evaporate that had stuck to the flask wall, and the concentrated extract was distributed among smaller flasks, which were left open in an exhaust hood for final evaporation and further concentration of the extract. After cooling, the material was filtered through No. 4 Whatman filter paper.

### 4.3. Extraction and Isolation

Initially, the ethanol extract was fractionated according to Tewierik et al. (2006) [30]. It was resuspended in 50 mL of a methanol/water solution (1:1) for 24 h, and was then subjected to a liquid/liquid partition, using solvents of increasing polarities in the order hexane, chloroform, and ethyl acetate. Sequentially, the fractions were analyzed by gas chromatography coupled to a mass spectrometer (GC–MS). Considering the results of this analysis, the isolation of the main compounds was carried out through separation by column chromatography, using silica gel as a stationary phase and hexane/ethyl acetate (90:10, 90:20, 85:15, and 80:20) as the mobile phase.

Next, the isolated compounds were characterized by gas chromatography coupled to a mass spectrometer (GC-MS), in a Shimadzu/GC 2010 device with a Shimadzu/AOC-500 auto-injector and a MS2010 plus mass detector with electronic impact ionization (IE, 70 and V), equipped with a DB-5MS fused silica column (Agilent J&W Advances 30 m × 0.25 mm × 0.25 µm), with helium at 65 kPa as the carrier gas. The conditions were as follows: oven temperature started at 100 °C for 2 min, increased to 290 °C at 4 °C min^−1^ and held for 6 min; injector and interface temperature was maintained at 250 °C; splitless 1 µL injection; helium was used as the carrier gas at a constant flow 0.75 mL min^−1^, and the run time was 30 min. The scan range was *m*/*z* 80–400. Retention times was 33 min. The final characterization was performed by infrared spectroscopy, using a Shimadzu IRAffinity-1 spectrometer operating with Fourier transform.

### 4.4. Cell Culture

Rat C6 glioblastoma cells (CCL-107™, ATCC) were cultured in DMEM supplemented with 10% fetal bovine serum and maintained in an incubator at 37 °C with 5% CO_2_, with the medium changed every 2 days. To perform the experiments, cells were washed with PBS and dissociated with 0.05% trypsin-EDTA, followed by centrifugation (1500 rpm for 3 min). After counting, cells were resuspended in culture medium and used to carry out the proposed assays.

Primary cultures of Müller cells were obtained from 9-day-old embryos. Retinal tissue was collected, and cells were distributed in 24-well plates (1 × 10^6^ cells/well). Cultures were grown for 10 days in DMEM supplemented with 10% fetal bovine serum and kept in an incubator at 37 °C with 5% CO_2_.

### 4.5. Cell Cytotoxicity Assay

Cells were plated (2 × 10^4^ cells/well) in 96-well plates and treated with eleutherine at different times and concentrations (0.5–50 μM for 6 h and 12 h, and 0.025–50 μM for 24 h). After the treatment period, the cells were incubated for 2 h with 0.5 mg/mL MTT solution (thiazolyl blue tetrazolium bromide—M2128, Sigma-Aldrich, St. Louis, MO, USA) in serum-free DMEM. Absorbance was measured with a microplate reader (BioRad, Hercules, CA, USA) at 570 nm. The results obtained were plotted with the mean and standard deviation of samples expressed as a percentage of the control value. We used GraphPad Prism 9 software to calculate the IC_50_.

### 4.6. Morphological Analysis of Eleutherine-Treated C6 Glioma Cells

Morphological alterations of C6 glioma cells were examined with a phase-contrast microscope (Leica DMI6000B). Cells were seeded in 12-well plate and incubated for 12 h with eleutherine at different concentrations (1 µM, 20 µM, 40 µM, 100 µM). Images were taken using a 63× objective.

### 4.7. Wound Healing/Scratch Migration Assay

A 12-well plate cell migration assay was performed to estimate the effect of eleutherine on C6 migration capacity. After 12 h of cell culture, a transverse lesion was made in the central surface of each well. Cells were treated with eleutherine at different concentrations (1 µM, 20 µM, 40 µM, and 100 µM). Cell migration to the injured area was followed by image capture using a digital camera coupled to an inverted microscope (Leica DMI6000B), using a 10× objective, at 0, 12, 24, and 48 h of treatment.

### 4.8. Colony Formation Assay

A colony formation assay was performed to evaluate the effect of eleutherine on cell proliferation. The C6 cells were used to seed a 12-well plate. After 24 h of culture, they were treated with eleutherine (1, 20, 40, and 100 µM) for 12 h. Cells were washed with PBS and dissociated with trypsin–EDTA (700 µL, 5 min). Quantification was made in a Neubauer chamber (1:1) in a 0.4% trypan blue solution. An aliquot of 1000 viable cells from each group was used to seed a 6-well plate and cultured for 7 days, with the medium changed every 2 days. Cultures were then washed with PBS and stained with 0.25% crystal violet/50% ethanol for 30 min.

### 4.9. Annexin V-FITC–Propidium Iodide (PI) Assay

Apoptosis/necrosis analysis was performed with the annexin V–FITC kit (Invitrogen). Cells were cultivated in a 12-well plate and treated with eleutherine (1, 20, and 40 µM) for 12 h. After that, cells were treated according to the manufacturer’s protocol. In each sample, 1 μL of annexin V–FITC and 2.5 μL of PI (250 μg/mL) were added and incubated for 10 min in an ice bath protected from light. The volume was made up to 250 μL with binding buffer and analyzed using a flow cytometer (BD FACSCanto II). Camptothecin (Sigma-Aldrich) was used as positive control (5 µM). The samples were analyzed using the flow cytometer with BD FACSDiva software, and a total of 10,000 events were collected for each sample and analyzed using the Flowing Software 2.5.1 (Turku, Finland). The final results were analyzed by cell percentage in each quadrant.

### 4.10. Western Blotting

The C6 cells were treated with eleutherine (1, 20, and 40 μM) for 12 h, after which cell lysates were prepared. Each lysate (60 µg of protein) was submitted to 10% SDS-PAGE gel electrophoresis. Gels were electrotransferred to nitrocellulose membranes (Hybond ECL, GE Healthcare, Uppsala, Sweden) by use of a membrane transfer system (Bio-Rad). Membranes were blocked with 5% skim milk at room temperature for 1 h, followed by overnight incubation with primary antibodies anti-AKT (cat.no.9272, Cell Signaling Technology) and anti-phospho-AKT (Ser473) (cat.no.9271, Cell Signaling Technology, Danvers, MA, USA) in 1:1000 dilution, and anti-GAPDH (C terminus) (SAB2500450, Sigma-Aldrich), dilution 1:500. After washing, the membranes were incubated with the secondary antibodies goat anti-mouse IgG, peroxidase conjugated in 1:2500 dilution, and horseradish peroxidase (HRP)-conjugated goat anti-rabbit IgG antibody used at 1:2500 (NIF824, GE Healthcare) for 1 h. Protein expression was detected using a chemiluminescent substrate kit (Merck Millipore, Burlington, MA, USA), and the images were captured by a ChemiDoc system (Bio-Rad, Hercules, CA, USA).

### 4.11. Real-Time Quantitative PCR Assay

Total RNA was extracted with TRI reagent (Applied Biosystems, Waltham, MA, USA), following the manufacturer’s instructions. The RNA concentration and quality were determined using a NanoDrop spectrophotometer (Kisker Biotech, Steinfurt, Germany) and 1% agarose gels. Complementary DNA was synthesized using high-capacity cDNA archive (Applied Biosystems, Waltham, MA, USA). The TERT mRNA expression (Rn01409457_m1) was evaluated by quantitative reverse transcription PCR (qRT-PCR) with primers and TaqMan probes purchased as Assays-on-Demand Products for Gene Expression (Applied Biosystems, Waltham, MA, USA). The GAPDH gene (Rn01462662_g1) was selected as an internal control for RNA input and reverse transcription efficiency. All real-time qRT-PCR reactions were performed in triplicate for the hTERT and GAPDH genes. Data were previously analyzed with the 2−ΔCt method and subsequently by relative quantification of the genes’ expression, which was calculated according to Livak and Schmittgen (2001) and Arocho et al. (2006) [49,50].

### 4.12. Ethics Statement

This study was approved by the Committee of Ethics of Animal Experiments of the Federal University of Pará (CEUA/UFPA 9381260919). We followed the guidelines found in the NIH Guide for the Care and Use of Laboratory Animals, and the experiments were carried out in compliance with the National Council for the Control of Animal Experimentation (CONCEA, Brazil).

### 4.13. Statistical Analysis

The data are expressed as the means ± SEM. The ANOVA followed by Tukey’s post hoc method using the statistical program GraphPad Prism 9. The significance levels are indicated as * *p* < 0.05, *** *p* < 0.01, and **** *p* < 0.0001.

## 5. Conclusions

In conclusion, our results suggest that eleutherine has cytotoxic and antiproliferative activity in glioma cells and induces their death by inhibiting the PI3K/AKT/telomerase pathway. This opens the important possibility that eleutherine is a viable cancer treatment, which should be further tested and studied in an in vivo model.

## Figures and Tables

**Figure 1 molecules-27-08850-f001:**
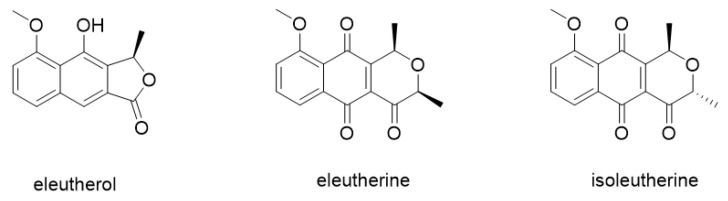
Chemical structures of eleutherol, eleutherine, and isoeleutherine.

**Figure 2 molecules-27-08850-f002:**
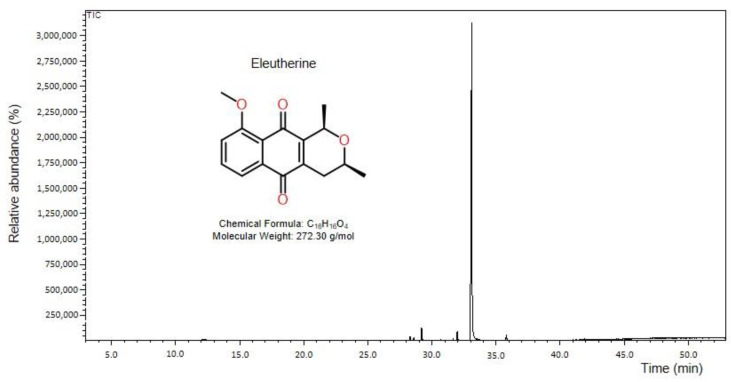
Chromatogram of eleutherine isolated from the bulbs of *Eleutherine plicata*. Peak area (95.03%) and retention time at 33.10 min.

**Figure 3 molecules-27-08850-f003:**
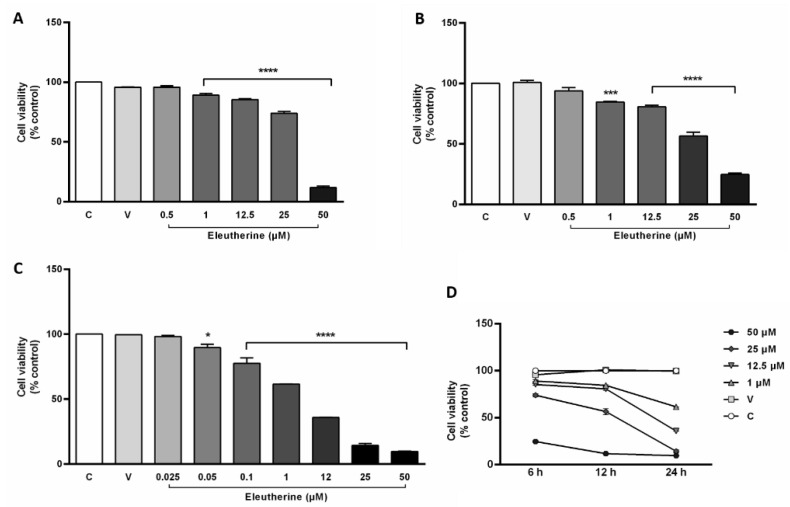
Eleutherine reduced the viability of glioblastoma cells as measured by the MTT assay. Cells were treated with different concentrations of eleutherine for 6 h (**A**), 12 h (**B**), and 24 h (**C**). (**D**) Cell viability curve showing the concentration and time-dependence of eleutherine’s effect. Data are presented as the mean ± SEM of three independent trials. Here, * *p* < 0.05, *** *p* < 0.001, and **** *p* < 0.0001 vs. untreated control (ANOVA, Tukey’s post hoc test); C, control; V, vehicle.

**Figure 4 molecules-27-08850-f004:**
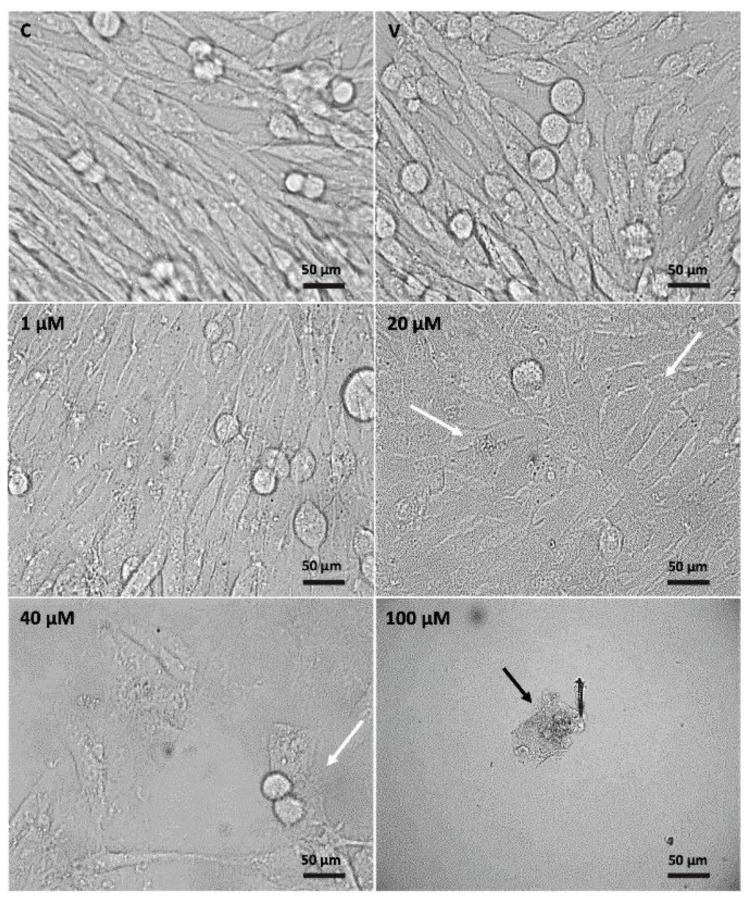
Morphological alterations in untreated C6 cells and after treatment with eleutherine at different concentrations (1, 20, 40, and 100 µM) for 12 h. Cells were examined using a phase-contrast microscope (Leica DMI6000B). White arrow indicates irregularly shaped cells with cytoplasmic shrinkage and black arrow indicates cell death. Images were taken using a 63× objective; C, control; V, vehicle.

**Figure 5 molecules-27-08850-f005:**
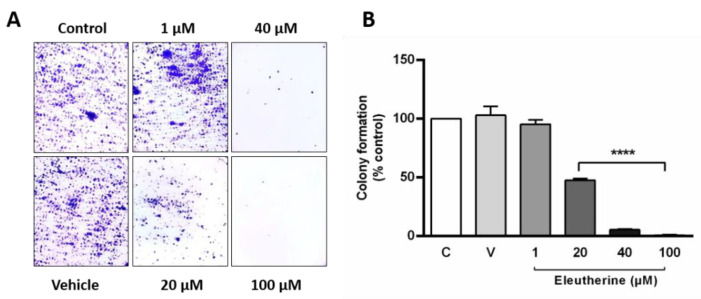
Eleutherine reduces colony formation in C6 cells. (**A**) C6 cells stained with crystal violet after eleutherine treatment for 12 h. (**B**) Graphic quantification of the colony numbers. Data are presented as the mean ± SEM of three independent trials. Here, **** *p* < 0.0001 vs. untreated control (ANOVA, Tukey’s post hoc test).

**Figure 6 molecules-27-08850-f006:**
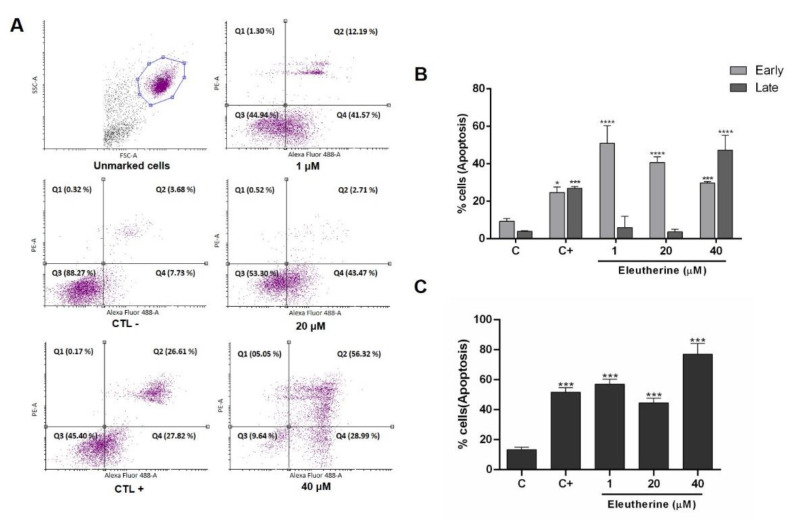
Eleutherine induces apoptosis in C6 cells. The cells were treated with eleutherine (1, 20, and 40 µM) for 24 h. (**A**) Cells were stained with annexin V–FITC and propidium iodide (PI) and subjected to flow cytometry analyses. Early apoptotic cells were in the Q4 quadrant and late apoptotic cells were in the Q2 quadrant. (**B**) Quantification of the percentage of early and late apoptotic C6 cells. (**C**) Quantification of the percentage of total apoptotic C6 cells. Here, * *p* < 0.05, *** *p* < 0.01, **** *p* < 0.0001 compared with untreated control (ANOVA, Tukey’s post hoc test); C, control; C+, positive control (camptothecin at 5 µM).

**Figure 7 molecules-27-08850-f007:**
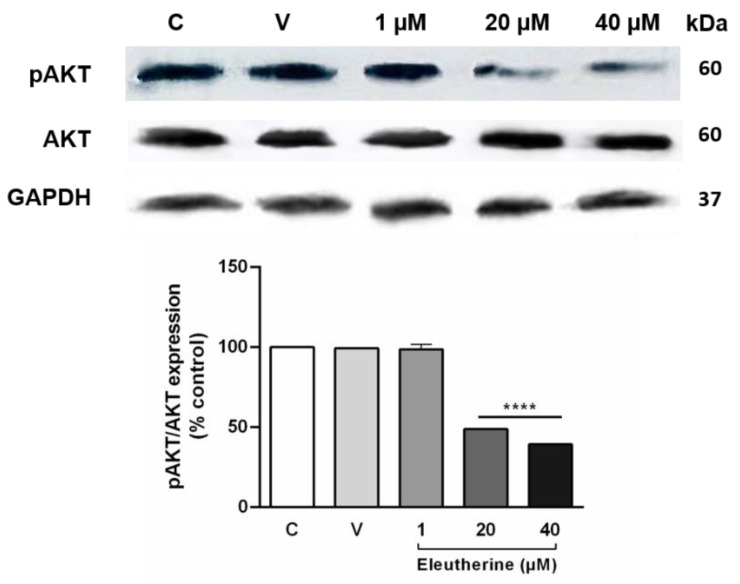
Eleutherine treatment reduced AKT phosphorylation in C6 cells. The expression of pAKT and AKT, and of GAPDH as the control protein, was detected by western blotting. The pAKT:AKT ratio is expressed as a percentage. The eleutherine treatment groups (20 and 40 µM) showed a greater reduction in AKT phosphorylation than was observed in the untreated group. Data are presented as the mean ± SEM of three independent trials. Here, **** *p* < 0.0001 vs. untreated control (ANOVA, Tukey’s post hoc test); C, control; V, vehicle.

**Figure 8 molecules-27-08850-f008:**
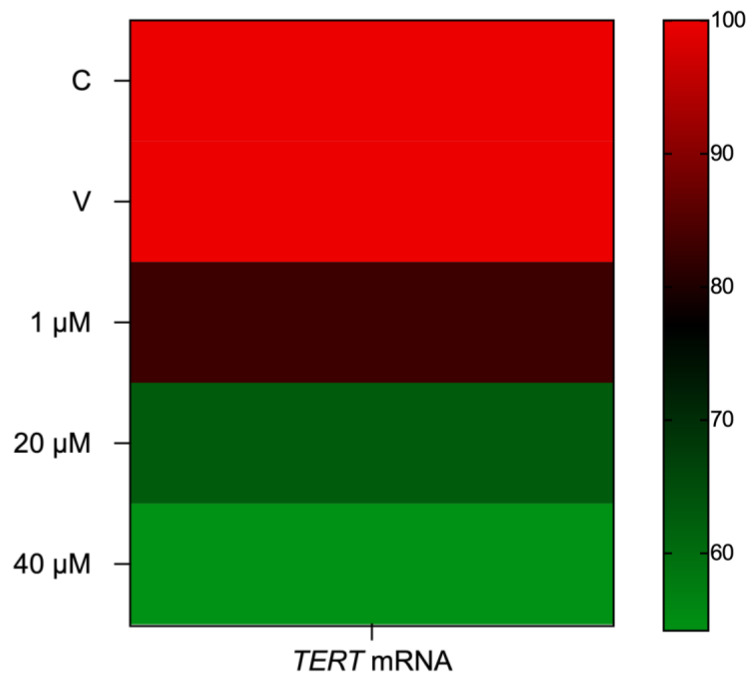
Heat map correlating eleutherine treatment with *TERT* mRNA expression in glioma cells. The color of each rectangle indicates the value of the Pearson correlation coefficient (ρ) between the eleutherine-treated groups and *TERT* mRNA expression.

## Data Availability

Not applicable.

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
