# Peer review of "In Vitro Cytotoxic Effects and Mechanisms of Action of Eleutherine Isolated from Eleutherine plicata Bulb in Rat Glioma C6 Cells"

_molecules, 2022, doi:10.3390/molecules27248850_

Round 1
Reviewer 1 Report
GENERAL COMMENTS:
The revised manuscript presents the cytotoxic effect of eleutherine in rat glioma cells and propose its possible mechanisms of action. If well, the manuscript represents and original study about the plant derived compound in the glioma treatment, there are not enough evidence in the results chapter to support the conclusion… Following, some comments are enumerated to improve a future presentation:
1. In the presented manuscript, only 1 cell line (C6) it is used to study the biological effects of eleutherine. At least 3 different glioma cell lines must be used to support the conclusions.
2. Moreover, to propose its use as anticancer specific compound, at least 1 normal/no tumoral glial cell line must to be used.
3. When the biological effects induced by eleutherine are studied, specifically in the migration assays, treatments need to be performed at no cytotoxic concentrations. If cytotoxic concentrations are used, the cell number and density alter the cell migration measured effects.
4. In addition, to support the eleutherine proapoptotic effect, only a cytometry assay was performed. More evidence with different experimental approaches as western blot, DNA laddering are indispensable to support the proapoptotic effect.
5. When the PI3K/AKT pathway is studied, only changes in pAKT are presented. Other proteins of the same pathway need to be analyzed to support the conclusions.
6. In relation to the previous comment, some explanation it is necessary to connect the “no dose response” changes in pAKT with the “dose response” changes observed in the mRNA telomerase expression studied by qRT-PCR. May be other methods to quantify telomerase changes can be applied.
PARTICULAR COMMENTS: Some punctual comments are listed below.
1. Line 2… The manuscript there is not study the antitumor effects… When the antitumor word is used, it is supposed that the data provide from animal models of research. In the current case, the word to use is “Citotoxic”.
2. Line 30… “inflammation, and tumor invasion, and promoting tumor regression.” Eliminate the “and” previous to “invasion”.
3. Line 64… Cites 18 to 21 did not are related to Eleutherine spp extracts. The current presentations of the paragraph is confusing. Please, reorder the redaction of contents…
4. Line 119… Revise the percentage values of colony formation. (…0.82%?).
5. Line 124… The legend of figure 4 must indicate what the white and black arrows means.
6. Line 132… “Eleutherine reduces C6 cell migration” The presented results are not valid to evaluate the migratory capacity. In the experiment, eleutherine was assayed at concentrations in which cell viability is affected (see figure 3). In consequence, it is not possible to assign the effect showed in figure 6 to the cell migration changes. In addition, the figure is not clear and have not a proper quality. I suggest removing this group of results.
7. Line 186… To change Eleutherine plicata to italics letters.
IN CONCLUSION, the revised manuscript is a very interesting work about the use of eleutherine in cancer, particularly in gliomas. However, in accordance to this reviewer criteria, it is necessary more evidence to support the conclusions. Summarizing, in accordance to the criteria of this reviewer, the manuscript it is not recommendable for publication in Molecules.
Author Response
GENERAL COMMENTS:
The revised manuscript presents the cytotoxic effect of eleutherine in rat glioma cells and propose its possible mechanisms of action. If well, the manuscript represents and original study about the plant derived compound in the glioma treatment, there are not enough evidence in the results chapter to support the conclusion… Following, some comments are enumerated to improve a future presentation:
Reply: We thank the reviewer comments towards our work and the opportunity to revise it and to make it clear.
- In the presented manuscript, only 1 cell line (C6) it is used to study the biological effects of eleutherine. At least 3 different glioma cell lines must be used to support the conclusions.
Reply: In this work, we focus primarily in glioma cell line (C6). This lineage has long been used as a glioma model for drug testing, because of the good correlation between cell lineage and cell fate (apoptosis) when drugs are tested. In addition, there are articles published by Molecules with only one lineage.
Chernov, A.N.; Filatenkova, T.A.; Glushakov, R.I.; Buntovskaya, A.S.; Alaverdian, D.A.; Tsapieva, A.N.; Kim, A.V.; Fedorov, E.V.; Skliar, S.S.; Matsko, M.M; Galimova, E.S.; Shamova, O.V. Anticancer Effect of Cathelicidin LL-37, Protegrin PG-1, Nerve Growth Factor NGF, and Temozolomide: Impact on the Mitochondrial Metabolism, Clonogenic Potential, and Migration of Human U251 Glioma Cells. Molecules, 2022, 27, 4988.
Othman, N.S.; Azman, D.K.M. Andrographolide Induces G2/M Cell Cycle Arrest and Apoptosis in Human Glioblastoma DBTRG-05MG Cell Line via
ERK1/2 /c-Myc/p53 Signaling Pathway. Molecules, 2022, 27,6686.
Gasparello, J.; Corradini, R.; Papi, C.; Zurlo, M.; Gambari, L.; Rozzi, A.; Manicardi, A.; Gambari, R.; Finotti, A. Treatment of Human Glioblastoma U251 Cells with Sulforaphane and a Peptide Nucleic Acid (PNA) Targeting miR-15b-5p: Synergistic Effects on Induction of Apoptosis. Molecules, 2022, 27,1299.
Yoo, K.; Yun, H.H.; Jung, S.Y.; Lee, J.H. KRIBB11 Induces Apoptosis in A172 Glioblastoma Cells via MULE-Dependent Degradation of MCL-1. Molecules, 2021, 26, 4165.
- Moreover, to propose its use as anticancer specific compound, at least 1 normal/no tumoral glial cell line must to be used.
Reply: This is a nice suggestion, we decided to include the figure as supplementary information.
S3: Eleutherine is nontoxic to nonneoplastic (glial) cells as analyzed by MTT assay. Cells were treated with different concentrations of eleutherine for 24 h. Data are presented as the mean ± SEM of three independent trials (ANOVA, Tukey post test). C, control; V, vehicle.
- When the biological effects induced by eleutherine are studied, specifically in the migration assays, treatments need to be performed at no cytotoxic concentrations. If cytotoxic concentrations are used, the cell number and density alter the cell migration measured effects.
Reply: We understand the reviewer's concern. We decided to follow the reviewer's suggestion and removed the migration assay data. In addition, we followed the reviewer's suggestion and performed the migration assay at non-cytotoxic concentrations to verify the parameters of normality, as shown in below figure.
- In addition, to support the eleutherine proapoptotic effect, only a cytometry assay was performed. More evidence with different experimental approaches as western blot, DNA laddering are indispensable to support the proapoptotic effect.
Reply: We have also looked to the question asked by the reviewer regarding the use of cytometry assay. In this method you can show the progression of the cells from viable (annexin V FITC negative, PI negative), to annexin FITC positive, PI negative to annexin V FITC positive, PI positive (dead cell). Furthermore, the major advantages of flow cytometry include the possibility of multiparameter measurements, correlation of different cellular events at a time, single cell analysis and rapid analysis of cell populations. This method overcomes a frequent problem of traditional bulk techniques such as fluorimetry, spectrophotometry, or gel techniques (Western blot) for allows correlative studies between many cell attributes based on both light scatter and fluorescence measurements. Therefore, we consider that cytometry assay is the most efficient and effective method of detection of apoptosis, including early, mid and late cell events, and can be detected, mainly for this detection assay. Also, flow cytometry can be employed to determine alterations in cell size. In addition, there are articles published by Molecules with just this assay.
Zid, D.A.; Saada, M.C.; Moslah, W.; Cartereau, M.P.; Lemettre, A.; Othman, H.; Gaysinski, M.; Koubaa, Z.A.; Souid, S.; Marrakchi, N.; Vandier, C.; Benkhadir, K.E.; Abid, N.S. AaTs-1: A Tetrapeptide from Androctonus australis Scorpion Venom, Inhibiting U87 Glioblastoma Cells Proliferation by p53 and FPRL-1 Up-Regulations. Molecules, 2021, 26, 7610.
Rodríguez, M.H.; Sánchez, P.I.M.; Martínez, J.; Pérez, M.E.M.; Cruz, E.R.; Zołek, T. Maciejewska, D.; Ruvalcaba, R.M.; Jiménez, E.M.; Vázquez, M.I.N. In Vitro and Computational Studies of Perezone and Perezone Angelate as Potential Anti-Glioblastoma Multiforme Agents. Molecules, 2022, 27, 1565.
Wlodkowic, D.; Telford, W.; Skommer, J.; Darzynkiewicz, Z. Apoptosis and Beyond: Cytometry in Studies of Programmed Cell Death. Methods Cell Biol, 2011, 103, 55–98.
Also we have performed an approach to identify the morphological alterations associated with apoptosis. Cell death was analyzed by phase contrast microscopy. Glioma cells treated with eleutherine decrease cell size, presence of morphological changes, such as cell retraction and cytoplasmic shrinkage characteristic of cells in process of apoptosis.
- When the PI3K/AKT pathway is studied, only changes in pAKT are presented. Other proteins of the same pathway need to be analyzed to support the conclusions.
We have also looked to the question asked by the reviewer regarding only changes in pAKT are presented, and we have shown that this pathway acts on glioma cells in a canonical way and may propagate signals through the phosphorylation and coordinate regulation of a variety of apoptotic regulators. PI3/AKT pathway is overexpressed in different cancer cell types and stimulation of this signaling pathway shows a critical function in cell growth, proliferation and survival in different cancer cells, especially in glioma cells, and it is related to direct regulation of hTERT expression and telomerase enzymatic activity by pAKT. Activation or suppression of telomerase/TERT can modulate cell cycle and induction of cell proliferation or apoptosis and senescence, respectively, and it is related to post-transcriptional regulation by phosphorylation of TERT by pAKT. Eleutherine could induce down regulation of hTERT expression by silencing AKT/pAKT, in the same pathway. The AKT pathway is the most commonly activated pathway in human cancers making it an attractive anticancer therapeutic target.
Wu, H.; Wei, M.; Li, Y.; Ma, Q.; Zhang, H. Research Progress on the Regulation Mechanism of Key Signal Pathways Affecting the Prognosis of Glioma. Front. Mol. Neurosci., 2022, 15, 910543.
Peek, G.W.; Tollefsbol, T.O. Down-regulation of hTERT and Cyclin D1 Transcription via PI3K/Akt and TGF-β Pathways in MCF-7 Cancer Cells with PX-866 and Raloxifene. Exp Cell Res, 2016, 344, 95-102.
Wu, X.Q; Huang, C.; He, X.; Tian, Y.Y.; Zhou, D.X.; He, Y.; Liu, X.H.; Li, J. Feedback regulation of telomerase reverse transcriptase: New insight into the evolving field of telomerase in cancer. Cellular Signalling, 2013, 25, 2462-2468.
Dratwa, M.; Wysoczanska, B.; Łacina, P.; Kubik, T.; Bogunia-Kubik, K. TERT Regulation and Roles in Cancer Formation. Front Immunol, 2020, 11, 589929.
Ghareghomi, S.; Ahmadian, S.; Zarghami, N.; Kahroba, H. Fundamental insights into the interaction between telomerase/TERT and intracellular signaling pathways. Biochimie, 2021, 181, 12-24.
- In relation to the previous comment, some explanation it is necessary to connect the “no dose response” changes in pAKT with the “dose response” changes observed in the mRNA telomerase expression studied by qRT-PCR. May be other methods to quantify telomerase changes can be applied.
Reply: We are grateful for the reviewer's comments on this interesting question.
The AKT pathway is important in the genesis of several types of cancer, being overexpressed and playing critical roles in the survival, proliferation, invasion, and migration of cancer cells. In our study, we found that eleutherine decreased the expression of p-Akt and inhibited the AKT/pAKT pathway, resulting in glioblastoma cell death. No evident dose-response relationship between expression of AKT/pAKT and TERT expression. Although there is not a dose response and no linear relationships found between the TERT expression and AKT/pAKT expression, these results suggest that inhibition detected by the immunoblotting in relation to the hTERT by the qRT-PCR may vary with differences in the methods used to measure them. However, the mechanism responsible for the discrepancies between the protein expression (AKT/pAKT) and expression of TERT remains unclear. Furthermore, it is important to consider that the apoptosis process does not occur homogeneously or synchronized in cells in the same culture group.
Regarding the assay used, we can say that qRT-PCR is the gold standard method for evaluating hTERT due to be affordable, reproducible and showed high specificity and sensibility approach for the analysis of mRNA expression comparing to other RNA analysis techniques.
PARTICULAR COMMENTS: Some punctual comments are listed below.
- Line 2… The manuscript there is not study the antitumor effects… When the antitumor word is used, it is supposed that the data provide from animal models of research. In the current case, the word to use is “Citotoxic”.
Reply: We thank the reviewer suggestion and now we provide modifications in tittle, since we received a comment from Reviewer #3. The new title, is:
“In vitro Cytotoxic Effects and Mechanisms of Action of Eleutherine Isolated from Eleutherine plicata bulb in Rat Glioma C6 Cells”
- Line 30… “inflammation, and tumor invasion, and promoting tumor regression.” Eliminate the “and” previous to “invasion”.
Reply: Done. The word was deleted.
- Line 64… Cites 18 to 21 did not are related to Eleutherine spp extracts. The current presentations of the paragraph is confusing. Please, reorder the redaction of contents…
Reply: We thank the reviewer suggestion and now we provide modifications. Two references were removed from paragraph. The new paragraph:
“Several plants, such as Eleutherine plicata, Eleutherine americana and Cipura paludosa, are important source of chemicals and contains high levels of naphthoquinones, a class of natural products with diverse biological activities. Aqueous extract of the dried bulbs of this these herbs have been described generates oxidative stress by inducing the deleterious endogenous formation of bioactive oxygen-derived species that inhibits inflammation, promotes apoptosis, and exhibits important anticancer properties in different cancer cell lines, such as glioma (U-251), breast (MCF-7), ovary (NCI/ADR-RES), kidney (786-0), lung non–small cell (NCI-H460), colon (HT-29), HepG2 cells, and leukemia (K562) [20-21].”
- Line 119… Revise the percentage values of colony formation. (…0.82%?).
Reply: The reviewer is correct. We did the revision.
- Line 124… The legend of figure 4 must indicate what the white and black arrows means.
Reply: We thank the reviewer for this advice, and we added the information in the legend.
- Line 132… “Eleutherine reduces C6 cell migration” The presented results are not valid to evaluate the migratory capacity. In the experiment, eleutherine was assayed at concentrations in which cell viability is affected (see figure 3). In consequence, it is not possible to assign the effect showed in figure 6 to the cell migration changes. In addition, the figure is not clear and have not a proper quality. I suggest removing this group of results.
Reply: We agree to the reviewer and removed the figure and the group of results.
- Line 186… To change Eleutherine plicata to italics letters.
Reply: Done.
IN CONCLUSION, the revised manuscript is a very interesting work about the use of eleutherine in cancer, particularly in gliomas. However, in accordance to this reviewer criteria, it is necessary more evidence to support the conclusions. Summarizing, in accordance to the criteria of this reviewer, the manuscript it is not recommendable for publication in Molecules.

Reviewer 2 Report
This is an interesting founding, I have some comment for this manuscript, as follow:
1. Figure 2. Please add the x-axis
2. Figure 7 A, the x and y axis are not clear enough
3. Figure 7. What concentration used for positive control?
4. Please add the information about structure determination in the supplementary
5. Please make the figure about the mechanism of eleutherine as anti-tumor
Please mention the specification of rat C6 glioblastoma cells used in this study.

Author Response
Reviewer #2
This is an interesting founding, I have some comment for this manuscript, as follow:
Reply: We thank the reviewer comments towards our work and the opportunity to revise it and to make it clear.
- Figure 2. Please add the x-axis.
Reply: Done. The x-axis was added.
- Figure 7 A, the x and y axis are not clear enough
Reply: Done. The x and y-axis were improved.
- Figure 7. What concentration used for positive control?
Reply: Camptothecin at 5 µM was used as positive control. The information was added in the legend.
- Please add the information about structure determination in the supplementary.
Reply: Two figures were added as supplementary. Two methods were used for structure determination, spectrum of mass (70 eV) and spectrum of FT-IR (KBr).
- Please make the figure about the mechanism of eleutherine as anti-tumor
Reply: We have added as graphical abstract.
Please mention the specification of rat C6 glioblastoma cells used in this study.
Reply: Rat C6 glioblastoma cells was obtained from ATCC (CCL-107™). The information was added in Materials and Methods, item 4.4.

Reviewer 3 Report
Manuscript Summary and Recommendation (ID: molecules-2039183)
Cancer remains one of the leading causes of mortalities worldwide. Though recent clinical advances have helped to increase survival rates, the incidence is still high and so is its mortality. Glioblastoma multiforme remain one of the most intransigent cancers. The prognosis, especially for high-grade gliomas, is dismal. Despite improvements in surgery techniques, radio‑ and chemotherapy, most patients present treatment resistance, recurrence and gliomas progression. Thus, the treatment of these aggressive primary brain tumors (of the central nervous system) represents one of the unmet needs in oncology. The submission ‘‘Antitumor Effects and Mechanisms of Action of Eleutherine in the Rat Glioma line C6 in vitro’’ by Shinkai et al. is an original contribution that addresses the foregoing need and is therefore potentially publishable in Molecules. Pyranonaphthoquinones and their derivatives (such as alpha and beta lapachone, ventiloquinone L, thysanone, nanaomycin A and eleutherin) investigated in this study are known to elicit various biological activities, including topoisomerase II inhibition and cytotoxicity on cancer cell lines. The authors should perform the following revisions to polish the current draft to a standard that is scientifically attractive to the readership of Molecules.
1. Title
Since isolation was performed, I suggest revising the title to: In vitro Antitumor Effects and Mechanisms of Action of Eleutherine Isolated from Eleutherine plicata bulb in Rat Glioma C6 Cells.
2. Abstract
L37-38: The abbreviations AKT/pAKT and qRT-PCR should be introduced in full at first mention. qRT-PCR and (TERT) could be omitted as they are not anywhere reused in the abstract.
3. Keywords
Should be revised, avoiding words already used in the title. I suggest adding Rat brain tumor model as an author-suggested indexing keyword.
4. Introduction
The medical challenge of cancer is worldwide. Therefore, I suggest putting this problem in a global perspective, giving an overview of the global cancer burden and what percentage gliomas take. After that, the text can conduct the reader to the information on gliomas (from L45 in the current draft).
L48: CNS should be expanded at first use.
L57: You may want to add more species (e.g., Cipura paludosa), because one species cannot be termed ‘‘Several plants’’.
5. Results
Section 2.1 should first report on the isolation and characterization of enantiopure Eleutherine from Eleutherine plicata bulbs. In this context, the authors should present clear data on the characterization using with GCMS for comparison with previously published data. The melting point was given under results, but the methodology does not contain any details pertaining to melting point determination. In the methodology, the authors cited using FTIR for characterization, but results of the same are not reported to support the GCMS data.
L95: IC50s?
6. Materials and methods
Fig. 10 is unnecessary, please omit it.
7. Conclusions
With the current order of presentation, this section should come before Materials and methods.
Author Response
Reviewer #3
Manuscript Summary and Recommendation (ID: molecules-2039183)
Cancer remains one of the leading causes of mortalities worldwide. Though recent clinical advances have helped to increase survival rates, the incidence is still high and so is its mortality. Glioblastoma multiforme remain one of the most intransigent cancers. The prognosis, especially for high-grade gliomas, is dismal. Despite improvements in surgery techniques, radio‑ and chemotherapy, most patients present treatment resistance, recurrence and gliomas progression. Thus, the treatment of these aggressive primary brain tumors (of the central nervous system) represents one of the unmet needs in oncology. The submission ‘‘Antitumor Effects and Mechanisms of Action of Eleutherine in the Rat Glioma line C6 in vitro’’ by Shinkai et al. is an original contribution that addresses the foregoing need and is therefore potentially publishable in Molecules. Pyranonaphthoquinones and their derivatives (such as alpha and beta lapachone, ventiloquinone L, thysanone, nanaomycin A and eleutherin) investigated in this study are known to elicit various biological activities, including topoisomerase II inhibition and cytotoxicity on cancer cell lines. The authors should perform the following revisions to polish the current draft to a standard that is scientifically attractive to the readership of Molecules.
Reply: We thank the reviewer comments towards our work and the opportunity to revise it and to make it clear.
- Title
Since isolation was performed, I suggest revising the title to: In vitro Antitumor Effects and Mechanisms of Action of Eleutherine Isolated from Eleutherine plicata bulb in Rat Glioma C6 Cells.
Reply: Done, partially. The Reviewer #1 suggested “cytotoxic effects”.
The new title is: “In vitro Cytotoxic Effects and Mechanisms of Action of Eleutherine Isolated from Eleutherine plicata bulb in Rat Glioma C6 Cells”
- Abstract
L37-38: The abbreviations AKT/pAKT and qRT-PCR should be introduced in full at first mention. qRT-PCR and (TERT) could be omitted as they are not anywhere reused in the abstract.
Reply: Done.
- Keywords
Should be revised, avoiding words already used in the title. I suggest adding Rat brain tumor model as an author-suggested indexing keyword.
Reply: Done. The keyword was added.
- Introduction
The medical challenge of cancer is worldwide. Therefore, I suggest putting this problem in a global perspective, giving an overview of the global cancer burden and what percentage gliomas take. After that, the text can conduct the reader to the information on gliomas (from L45 in the current draft).
Reply: Done. The paragraph was rewritten and new paragraph was added, with two news reference.
Former: Glioma is a fatal malignancy derived from the glial cell lineage. Although less prevalent than other neoplasms, 60% are aggressive high-grade gliomas (grade IV), which migrate and strongly infiltrate the brain parenchyma [1-5]. Gliomas, in general, are responsible for approximately 38% of primary brain tumors and represent more than 70% of all CNS tumors, the most frequent and malignant being glioblastoma multiforme [6,7].
New: Glioma is a fatal malignancy derived from the glial cell lineage with a high recurrence rate. Although less prevalent than other neoplasms, 60% are aggressive high-grade gliomas (grade IV), which migrate and strongly infiltrate the brain parenchyma [1-5]. Gliomas, in general, are responsible for approximately 38% of primary brain tumors and represent more than 70% of all Central Nervous System (CNS) tumors, the most frequent and malignant being glioblastoma multiforme [6,7]. Glioblastoma have the highest proportion of malignancy and the worst prognosis of CNS tumors. Only 5.5% of patients usually survive for 5 years after diagnosis (Ostrom et al., 2020) and have a one-year survival rate because of the limitations of treatment’s approaches (Fernandes et al., 2017).
L48: CNS should be expanded at first use.
Reply: Done.
L57: You may want to add more species (e.g., Cipura paludosa), because one species cannot be termed ‘‘Several plants’’.
Reply: Done. The new paragraph is: Several plants, such as Eleutherine plicata, Eleutherine americana and Cipura paludosa, contains high levels of naphthoquinones, a class of natural products with diverse biological activities.
- Results
Section 2.1 should first report on the isolation and characterization of enantiopure Eleutherine from Eleutherine plicata bulbs. In this context, the authors should present clear data on the characterization using with GCMS for comparison with previously published data. The melting point was given under results, but the methodology does not contain any details pertaining to melting point determination. In the methodology, the authors cited using FTIR for characterization, but results of the same are not reported to support the GCMS data.
Reply: To clarify the protocol, we added more information in the methods (in item 4.3.), in the results (item 2.1.) as follows, and were added two supplementary figures.
Item 4.3. “The conditions were: oven temperature started at 100 °C for 2 min, increased to 290 °C at 4 °C min-1 and held for 6 min; injector and interface temperature was maintained at 250 °C; splitless 1 µL injection; helium was used as the carrier gas at a constant flow 0.75 mL min-1, and the run time was 30 min. The scan range was m/z 80-400. Retention times for was 33 min.”
Item “2.1. Characterization of the compound isolate: The compound isolated from Eleutherine plicata bulbs was obtained as brownish yellow crystal. Starting from 3 g of the lyophilized ethanolic extract yield over 49 mg of the pure compound, with 95% relative concentration (Figure 2), with MS (70 eV, EI) 272 (42), 257 (100), 243 (52), 214 (32),157 (10) and 121 (25) and the probable molecular formula was deduced as C14H12O4. The melting point was of 144–146 °C. The IR spectrum showed the presence characteristic peaks, such as, -C-Hsp2 (2974 cm−1), -C-Hsp3 (2913), -C=O (1777 cm−1), of benzene ring (1582cm−1) absorptions. The molecular characterization data were compared with those in the literature [31].”
L95: IC50s?
Reply: We thank the reviewer for mistake identification. We omitted “s”.
- Materials and methods
Fig. 10 is unnecessary, please omit it.
Reply: Done. Figure 10 was removed.
- Conclusions
With the current order of presentation, this section should come before Materials and methods.
Reply: The reviewer is correct, but we following the Molecules template (available in https://www.mdpi.com/journal/molecules/instructions#preparation).

Round 2
Reviewer 1 Report
The corrected manuscript entitled “In vitro Cytotoxic Effects and Mechanisms of Action of Eleutherine Isolated from Eleutherine plicata bulb in Rat Glioma C6 Cells” evidence a significant improvement in relation to the revised previous version. Both, general and particular comments were attended and explained correctly by the authors. There are no additional comments to incorporate; in consequence, this reviewer suggest the publication of the manuscript.